# A Proposal of Mode Polynomials for Efficient Use of Component Mode Synthesis and Methodology to Simplify the Calculation of the Connecting Beams

Jeong Hee Park [1],* and Duck Young Yoon [2]

1   Structural Design Department, Hyundai Samho Heavy Industries, Jeollanam-do 58462, Korea
2   Department of Naval Architecture and Ocean Engineering, Chosun University, Gwangju 61452, Korea;
    dyyun@chosun.ac.kr
*   Correspondence: parkjh@hshi.co.kr

**Abstract:** Analytical method using Rayleigh–Ritz method has not been widely used recently due to intensive use of finite element analysis (FEA). However as long as suitable mode functions together with component mode synthesis (CMS) can be provided, Rayleigh–Ritz method is still useful for the vibration analysis of many local structures in a ship such as tanks and supports for an equipment. In this study, polynomials which combines a simple and a fixed support have been proposed for the satisfaction of boundary conditions at a junction. Higher order polynomials have been generated using those suggested by Bhat. Since higher order polynomials used only satisfy geometrical boundary conditions, two ways are tried. One neglects moment continuity and the other satisfies moment continuity by sum of mode polynomials. Numerical analysis have been performed for typical shapes, which can generate easily more complicated structures. Comparison with FEA result shows good agreements enough to be used for practical purpose. Frequently dynamic behavior of one specific subcomponent is more concerned. In this case suitable way to estimate dynamic and static coupling of subcomponents connected to this specific subcomponent should be provided, which is not easy task. Elimination of generalized coordinates for subcomponents by mode by mode satisfaction of boundary conditions has been proposed. These results are still very useful for initial guidance.

**Keywords:** mode polynomials; CMS (component mode synthesis); FEA (finite element analysis); PMSC (proposal of methodology for a simplification of computation)

## 1. Introduction

Each tank installed on the ship is arranged in the stern and engine room of the ship considering the cargo loading space, and there is a possibility that excessive vibration may occur due to the main excitation forces (main engine and propeller) that causes the ship vibration. If excessive vibration occurs after drying, it occurs significant restrictions and high cost on reinforcing work such as welding and special painting inside the tank. Therefore, anti-vibration measures are required at the design stage, along with a commercial program (MSC Patran/Nastran), in some cases, a calculation program [1,2] that can simply check the natural frequency has been developed and used.

In order to have an anti-vibration design of structures through calculation methods, it is necessary to analyze the normal mode for resonance avoidance with the main excitation force. For the normal mode analysis of structures, analytical methods such as Rayleigh–Ritz method are widely used together with finite element method. Although the finite element method is widely used in recent years, the analytical method which can be easily and simply reviewed at the initial design stage is still useful because the calculation time is longer than that of the analytical method.

Analytical methods require an approach that can yield more reliable results. In general, when the analytical method is applied, the Euler's beam function is used.

However, since the operation is very complicated, a study has been made on a polynomial having a beam property in order to simplify it. Park and Yang [3] performed the calculation of the natural frequency of the connected rectangular plate using a polynomial. Bhat [4] uses a polynomial as mode functions. Han [5,6] analyzed the complex vibration of the panel using the assumed mode method as an analytical method.

Kim [7] carried out the vibration analysis of the rectangular reinforced plate using polynomial with Timoshenko beam function property which can consider the rotational inertia and shear deformation effect of plate and stiffener. However, all calculations in the above mentioned studies are performed with given boundary conditions. The above mentioned studies were performed with differentiated boundary condition such as simple and fixed boundary conditions for a single structure.

However, the ship structure is not a single but a connection structure. Therefore, it is not appropriate to calculate the connection structure by simply assigning a simple or fixed boundary condition. CMS (component mode synthesis) method was applied to calculate the connection structure [8,9]. The first CMS method was presented by Hurty [10] in 1960. Alessandro Cammarata [11] introduced a wide range of CMS content, described an algorithm applied to flexible multi-objects, and a method of reducing degrees of freedom. In order to calculate the normal mode analysis of the connected structure using the CMS method, it is important to define the constraint at the connection part, and various studies on the constraint conditions at the junction were performed by Hurty et al. [12–18].

Carrera et al. [19] is developed theory that can be solved by converting a three-dimensional model for large deformation of a structure into one dimension was developed and applied to the calculation. Pagani et al. [20] is explained that the natural frequency and mode shape can be changed significantly when the metal structure is subjected to large displacement and rotation under geometrical nonlinear conditions.

Further, geometric nonlinear total Lagrange formula including cross-sectional deformation was developed to implement the vibration mode of the composite beam structure in the nonlinear region [21].

As mentioned above, many studies have been conducted, but it is still important to find a way to minimize the convergence of boundary conditions at the junction. This study proposed the following method to minimize the convergence of boundary conditions at the junction.

Firstly, we have proposed polynomials combining fixed and simple supports to satisfy boundary condition at junctions between each subsystem. We know that this approach has never been tried.

Secondly, although Bhat [4] proposed a fixed and simple support function, the calculation was performed by applying it to a simple plate. In addition, the function proposed by Bhat does not satisfy the natural condition in higher order terms of second or higher order.

In this study, in order to compensate for this problem, calculations were performed for the two cases mentioned below at the connection point and the results were compared in Section 4.1.

(1)   Displacement, slope, and moment continuity (total sum of natural conditions is continuous);
(2)   Displacement, and slope continuity (ignoring natural conditions).

For reference, the geometrical boundary condition mentioned in this manuscript refers to the boundary condition for displacement and slope, and the natural boundary condition refers to the boundary condition for moment. [4]

Third, in order to confirm the usefulness of the proposed method, a numerical analysis was performed on the representative shape of two and three components typical.

In particular, for the two component type, various verifications were performed in the entire length range $0 \ll x \ll 1$ according to the length ratio ($L_A$:$L_B$).

Fourth, frequently, only specific subcomponent is more concerned for vibration analysis. In this case, the suitable boundary conditions to consider the static and dynamic coupling from the other subcomponent through junctions should be provided. However, the suggestions for such boundary conditions are hardly found. In this study, in order to calculate the above case, A simplified method that can reduce the degree of freedom up to 50% by matching the subcomponent mode and the interest component mode as a constraint condition at the junctions is also proposed.

The purpose of the simplification method presented in this study is to show that it is possible to calculate a method that can reduce the degree of freedom by 50%, rather than a method for comparing numerical calculation results with the existing method. Although this method is somewhat excessive, as a result, it satisfies the finite element analysis (FEA) result and the analysis error of 15%, which is appropriate as an approximate numerical methodology, so it is considered to be efficient for approximate numerical calculation.

In addition, a three component structure was used for the calculation of structures in which symmetric and asymmetric modes occur repeatedly. A mode function having an appropriate boundary condition for a three component structure is proposed. The case of three component structures, fixed-fixed, fixed-simple, simple-fixed, fixed-free, simple-simple, and displacement functions were used.

A method of simplifying and calculating it for asymmetric and symmetric modes was proposed.

## 2. Definition of Assumed Mode Functions

The component mode synthesis is suitable for the vibration analysis of many local structures in ships such as tanks and supports for equipment, in which structures are divided into smaller subcomponents.

In the component mode synthesis, each mode function does not need to satisfy the junction conditions as long as their combined sum allows these junction conditions to be satisfied.

Nevertheless efficient mode functions to improve convergence are still very important for the practical use. Polynomials are frequently considered as mode functions [4]. Junction conditions among subcomponent are neither fixed nor simple support. It is a reasonable guess for mode functions to be represented by combined sum of functions for fixed and simple support. To represent the basic ideas of the method of modal synthesis, an example shown in Figure 1 is used. Vibration only in the plane of paper is considered.

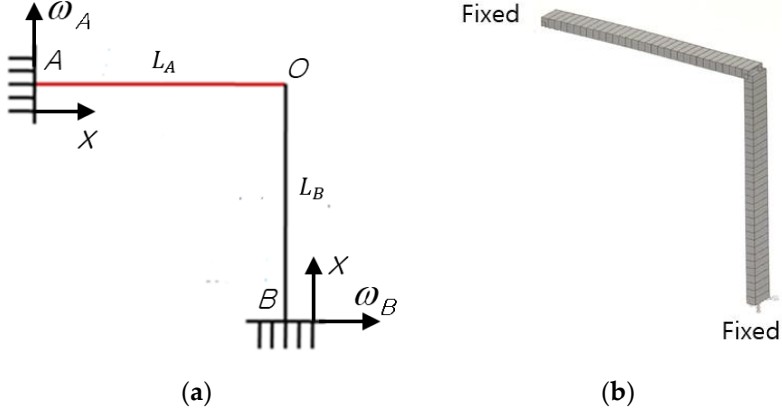

**(a)**                                                    **(b)**

**Figure 1.** Structure model of two section type connected beam. (**a**) Simplify model; (**b**) finite element analysis (FEA) Model.

### 2.1. Two Components Type Connected Structure

The beam is separated into two sections OA and OB, whose coordinates are shown as $w_A$; $x$ and $w_B$; $x$.

The properties of structures used in the numerical analysis are shown in Table 1.

**Table 1.** Properties and cross section of model.

| Category | $W_A$ | $W_B$ | Cross Section |
|---|---|---|---|
| Density [kg/m$^3$] | 7850 | 7850 | |
| Length [m] | 5 | 5/4/2.5 | |
| Area [m$^2$] | 0.1 | 0.1 | |
| Young's modulus [N/mm$^2$] | $2.10 \times 10^5$ | $2.10 \times 10^5$ | |
| 2nd moment of area [mm$^4$] | $3.33 \times 10^8$ | $3.33 \times 10^8$ | |

Deflections can be shown as below.

$$W_A(x,t) = \sum_{i=1}^{m} w_{Ai}(x) \cdot p_{Ai}(t) \tag{1}$$

$$W_B(x,t) = \sum_{i=1}^{n} w_{Bi}(x) \cdot p_{Bi}(t) \tag{2}$$

For the simple explanation of component mode synthesis, Euler beam are assumed and $(EI)_A = (EI)_B = EI$.

Where E and I are Young's modulus and 2nd moment of area, and $L_A$ and $L_B$ are length of beams.

The deflection of subcomponent OA and OB using fixed and simple boundary condition can be expressed as below.

Where $x_A$ and $x_B$ are non-dimensional such that $\zeta = \frac{x_A}{L_A}$, $\xi = \frac{x_B}{L_B}$:

$$W_A(\zeta,t) = \sum_{i=1}^{m} \left( \psi_i(\zeta) p_{Ai}(t) + \phi_i(\zeta) q_{Ai}(t) \right) \tag{3}$$

$$W_B(\xi,t) = \sum_{j=1}^{n} \left( \psi_j(\xi) p_{Bj}(t) + \phi_j(\xi) q_{Bj}(t) \right) \tag{4}$$

and $p_{Ai}(t), q_{Ai}(t), p_{Bj}(t), q_{Bj}(t)$ are the general coordinate system in the mode function of beam.

The polynomials mode function for $\psi_i$, $\phi_i$ are suggested such that $\psi_i$ for fixed-fixed boundary condition and $\phi_i$ for fixed-simple boundary conditions.

$\psi_1$ can be derived by assuming fourth order polynomial and boundary conditions.

Looking at the process of deriving a fixed-fixed function:

$$\psi_1(\zeta) = a_0 + a_1 \times \zeta + a_2 \times \zeta^2 + a_3 \times \zeta^3 + a_4 \times \zeta^4 \tag{5}$$

Applying geometric boundary condition $\psi_1(0) = 0$, $\psi_1{}'(0) = 0$, $\psi_1(1) = 0$ and $\psi_1{}'(1) = 0$ to Equation (5) yields a fixed-fixed fundamental mode function such as Equation (6).

$$\psi_1(\zeta) = \left( \zeta^4 - 2\zeta^3 + \zeta^2 \right) A_1 \tag{6}$$

$\phi_1$ can be derived by assuming 3rd order (only used geometric boundary condition) and fourth order (combine used geometric and natural boundary condition) polynomial and boundary conditions.

First, the fundamental mode function of a third polynomial is

$$\phi_1(\zeta) = a_0 + a_1 \times \zeta + a_2 \times \zeta^2 + a_3 \times \zeta^3 \tag{7}$$

Applying $\phi_1(0) = 0$, $\phi_1{}'(0) = 0$ and $\phi_1(1) = 0$ to Equation (7) yields a fixed-simple fundamental mode function such as Equation (8):

$$\phi_1(\zeta) = \left( \zeta^3 - \zeta^2 \right) B_1 \tag{8}$$

and the fundamental mode function of a fixed-simple 4th polynomial is

$$\phi_1(\zeta) = a_0 + a_1 \times \zeta + a_2 \times \zeta^2 + a_3 \times \zeta^3 + a_4 \times \zeta^4 \tag{9}$$

Applying $\phi_1(0) = 0$, $\phi_1{}'(0) = 0$, $\phi_1(1) = 0$ and $\phi_1{}''(1) = 0$ to Equation (9) yields a fixed-simple fundamental mode function such as Equation (10):

$$\phi_1(\zeta) = \zeta(\zeta - 1)\left(2\zeta^2 - 3\zeta\right)B_1 \tag{10}$$

The coefficients $A_1$ and $B_1$ are implemented using the orthogonal formula of the beam function:

$$\int_0^1 \psi_i \psi_j d\zeta = \delta_{ij} \tag{11}$$

$$\int_0^1 \phi_i \phi_j d\zeta = \delta_{ij} \tag{12}$$

where $i$, $j$ is the vibration order, and $\delta_{ij}$ is Kronecker delta.

$$A_1 = \frac{\int_0^1 \psi_1^2 d\zeta}{\sqrt{\int_0^1 \left(\zeta^4 - 2\zeta^3 + \zeta^2\right)^2 d\zeta}} \tag{13}$$

$$B_1 = \frac{\int_0^1 \phi_1^2 d\zeta}{\sqrt{\int_0^1 \left(\zeta^4 - 2.5\zeta^3 + 1.5\zeta^2\right)^2 d\zeta}} \tag{14}$$

and the mode function of the second mode or more can be implemented from the following Equations (15) to (16). The expansion to higher mode is the same regardless of the third or fourth fundamental mode function:

$$\psi_k = A_k\left[\psi_{k-1} \times \zeta - \sum_{i=1}^{k-1} a_{ki} \times \psi_{k-1}\right] \tag{15}$$

$$\phi_k = B_k\left[\phi_{k-1} \times \zeta - \sum_{i=1}^{k-1} b_{ki} \times \phi_{k-1}\right] \tag{16}$$

The coefficients $a_{ki}$, $b_{ki}$ can be obtained from the orthogonal relation of the Equations (15) to (16).

The mass and stiffness matrix was obtained for each of the defined mode functions, and slope was given as constraint at the connection point to implement the natural frequency and mode of the connected beams.

The mass matrix and stiffness matrix using suggested polynomials are shown in Equations (17) and (18):

$$[M_A] = mL_A \begin{vmatrix} \psi_1\psi_1 & \cdots & \psi_1\psi_m & \psi_1\phi_1 & \cdots & \psi_1\phi_m \\ \vdots & \ddots & \vdots & \vdots & & \vdots \\ \psi_m\psi_1 & \cdots & \psi_m\psi_m & \psi_m\phi_1 & \cdots & \psi_m\phi_m \\ \phi_1\psi_1 & \cdots & \phi_1\psi_m & \phi_1\phi_1 & \cdots & \varphi_1\phi_m \\ \vdots & \ddots & \vdots & \vdots & & \vdots \\ \phi_m\psi_1 & \cdots & \phi_m\psi_m & \phi_m\phi_1 & \cdots & \phi_m\phi_m \end{vmatrix} \tag{17}$$

$$[K_A] = \frac{8EI}{L_A^3} \begin{vmatrix} \psi_1'' \psi_1'' & \cdots & \psi_1'' \psi_m'' & \psi_1'' \phi_1'' & \cdots & \psi_1'' \phi_m'' \\ \vdots & \ddots & \vdots & \vdots & & \vdots \\ \psi_m'' \psi_1'' & \cdots & \psi_m'' \psi_m'' & \psi_m'' \phi_1'' & \cdots & \psi_m'' \phi_m'' \\ \phi_1'' \psi_1'' & \cdots & \phi_1'' \psi_m'' & \phi_1'' \phi_1'' & \cdots & \phi_1'' \phi_m'' \\ \vdots & \ddots & \vdots & \vdots & & \vdots \\ \phi_1'' \psi_1'' & \cdots & \phi_m'' \psi_m'' & \phi_m'' \varphi_1'' & \cdots & \phi_m'' \phi_m'' \end{vmatrix} \tag{18}$$

where $m$ is mass per unit length, we can express the mass matrix as follows by using orthogonality:

$$[M_A] = mL_A \begin{vmatrix} 1 & \cdots & 0 & \psi_1 \phi_1 & \cdots & \psi_1 \phi_m \\ \vdots & \ddots & \vdots & \vdots & & \vdots \\ 0 & \cdots & 1 & \psi_m \phi_1 & \cdots & \psi_m \phi_m \\ \phi_1 \psi_1 & \cdots & \phi_1 \psi_m & 1 & \cdots & 0 \\ \vdots & \ddots & \vdots & \vdots & \ddots & \vdots \\ \phi_m \psi_1 & \cdots & \phi_m \psi_m & 0 & \cdots & 1 \end{vmatrix} \tag{19}$$

and $M_B$, $K_B$ can be expressed in a similar.

Therefore, the mass and stiffness matrix of the subcomponents in Figure 1 can be expressed as in Equations (20) and (21):

$$[M] = \begin{vmatrix} M_A & 0 \\ 0 & M_B \end{vmatrix} \tag{20}$$

$$[K] = \begin{vmatrix} K_A & 0 \\ 0 & K_B \end{vmatrix} \tag{21}$$

Note that no coupling between the displacement of OA and that of OB.

Note that the number of generalized coordinates shall be 2 $(m + n)$, in order to simplify the calculation process, only slope was assigned as a constraint condition at the connection point.

The coordinate system reflecting the constraints is expressed in Equation (22); where $\alpha$ is the ratio of length for subcomponents ($\alpha = L_B/L_A$):

$$\begin{Bmatrix} p_{A1} \\ \vdots \\ p_{Am} \\ q_{A1} \\ \vdots \\ q_{Am} \\ p_{B1} \\ \vdots \\ p_{Bn} \\ q_{B1} \\ \vdots \\ q_{Bn} \end{Bmatrix} = \begin{vmatrix} 1 & \cdots & 0 & 0 & \cdots & 0 & 0 & \cdots & 0 & 0 & \cdots & 0 \\ \vdots & \ddots & \vdots & \vdots & \ddots & \vdots & \vdots & \ddots & \vdots & \vdots & \ddots & \vdots \\ 0 & \cdots & 1 & 0 & \cdots & 0 & 0 & \cdots & 0 & 0 & \cdots & 0 \\ 0 & \cdots & 0 & 1 & \cdots & 0 & 0 & \cdots & 0 & 0 & \cdots & 0 \\ \vdots & \ddots & \vdots & \vdots & \ddots & \vdots & \vdots & \ddots & \vdots & \vdots & \ddots & \vdots \\ 0 & \cdots & 0 & 0 & \cdots & 1 & 0 & \cdots & 0 & 0 & \cdots & 0 \\ 0 & \cdots & 0 & 0 & \cdots & 0 & 1 & \cdots & 0 & 0 & \cdots & 0 \\ \vdots & \ddots & \vdots & \vdots & \ddots & \vdots & \vdots & \ddots & \vdots & \vdots & \ddots & \vdots \\ 0 & \cdots & 0 & 0 & \cdots & 0 & 0 & \cdots & 1 & 0 & \cdots & 0 \\ 0 & \cdots & 0 & 0 & \cdots & 0 & 0 & \cdots & 0 & 1 & \cdots & 0 \\ \vdots & \ddots & \vdots & \vdots & \ddots & \vdots & \vdots & \ddots & \vdots & \vdots & \ddots & \vdots \\ 0 & \cdots & 0 & \alpha\frac{\phi_1'}{\phi_n'} & \cdots & \alpha\frac{\phi_m'}{\phi_n'} & 0 & \cdots & 0 & -\frac{\phi_1'}{\phi_n'} & \cdots & -\frac{\phi_{n-1}'}{\phi_n'} \end{vmatrix} \begin{Bmatrix} p_{A1} \\ \vdots \\ p_{Am} \\ q_{A1} \\ \vdots \\ q_{Am} \\ p_{B1} \\ \vdots \\ p_{Bn-1} \\ q_{B1} \\ \vdots \\ q_{Bn-1} \end{Bmatrix} \tag{22}$$

the $[M]$, $[K]$, and $[C]$ matrix obtained in this way were substituted into the Lagrange equation of motion to calculate the natural frequencies, and the results are mentioned in Section 4.

### 2.2. Three Components Type Connected Structure

The beam is separated into three sections AB, BD, and CD, whose coordinates are shown as $w_A; x, w_B; x$, and $w_C; x$.

The properties of structures used in the numerical analysis are shown in Table 2.

**Table 2.** Properties & cross section of model.

| Category | $W_A$ | $W_B$ | $W_C$ | Cross Section |
|:---:|:---:|:---:|:---:|:---:|
| Density [kg/m$^3$] | 7850 | 7850 | 7850 | |
| Length [m] | 5 | 5/2.5 | 5 | |
| Area [m$^2$] | 0.1 | 0.1 | 0.1 | |
| Young's modulus [N/mm$^2$] | $2.10 \times 10^5$ | $2.10 \times 10^5$ | $2.10 \times 10^5$ | |
| 2nd moment of area [mm$^4$] | $3.33 \times 10^8$ | $3.33 \times 10^8$ | $3.33 \times 10^8$ | |

Deflections can be shown as below

$$W_A(x,t) = \sum_{i=1}^{m} w_{Ai}(x) \cdot p_{Ai}(t) \tag{23}$$

$$W_B(x,t) = \sum_{i=1}^{n} w_{Bi}(x) \cdot p_{Bi}(t) \tag{24}$$

$$W_C(x,t) = \sum_{i=1}^{k} w_{Ci}(x) \cdot p_{Ci}(t) \tag{25}$$

The deflection of subcomponent *AB*, *CD* and *BD* using fixed and simple boundary condition can be expressed as below:

$$W_A(\zeta,t) = \sum_{i=1}^{m} (\psi_i(\zeta)p_{Ai}(t) + \phi_i(\zeta)q_{Ai}(t) + \gamma_i(\zeta)r_{Ai}(t)) \tag{26}$$

$$W_B(\xi,t) = \sum_{j=1}^{n} (\phi_j(\xi)q_{Bj}(t) + v_j(\xi)s_{Bj}(t) + \lambda_j(\xi)o_{Bj}(t)) \tag{27}$$

$$W_C(\eta,t) = \sum_{k=1}^{s} (\psi_k(\eta)p_{Ck}(t) + \phi_k(\eta)q_{Ck}(t) + \gamma_k(\eta)r_{Ck}(t)) \tag{28}$$

$$U_{B0}(\xi,t) = u_{b0}(t) \tag{29}$$

$$U_{B1}(\xi,t) = u_{b1}(t) \tag{30}$$

where $x_A$, $x_B$, and $x_c$ are non-dimensional such that $\zeta = \frac{x_A}{L_A}$, $\xi = \frac{x_B}{L_B}$, $\eta = \frac{x_C}{L_C}$. $p_{Ai}(t), q_{Ai}(t)$, $r_{Ai}(t), q_{Bj}(t), s_{Bj}(t), o_{Bj}(t), p_{Ck}(t), q_{Ck}(t), r_{Ck}(t), u_{b0}(t), u_{b1}(t)$ are the general coordinate system in the mode function of beams, and $U_{B0}$ and $U_{B1}$ are displacement functions of both sides in horizontal beam. $L_A$, $L_B$ and $L_C$ are length of beams, separately.

Polynomial mode functions $\psi_i$ and $\phi_i$ are mentioned in Section 2.1. Where $\gamma_i$, $v_j$, and $\lambda_j$ are functions for fixed-free, simple-fixed, and simple-simple boundary conditions, respectively.

The fundamental mode function reflecting each boundary condition is shown in Tables 3–5.

**Table 3.** Fixed-free fundamental mode function ($\gamma_1$).

| Boundary Condition | Mathematical Expression | Fundamental Mode Function |
|---|---|---|
| GBC (2) + NBC (2) | $\gamma_1(0) = 0,\ \gamma_1'(0) = 0$ <br> $\gamma_1''(1) = 0,\ \gamma_1'''(1) = 0$ | $\zeta^4 - 4\zeta^3 + 6\zeta^2$ |

GBC (2): Geometric Boundary Condition (number of boundary condition). NBC (2): Natural Boundary Condition (number of boundary condition).

**Table 4.** Simple-fixed fundamental mode function ($v_1$).

| Boundary Condition | Mathematical Expression | Fundamental Mode Function |
|---|---|---|
| GBC (3) + NBC (1) | $v_1''(0) = 0,\ v_1(0) = 0$ <br> $v_1(1) = 0,\ v_1'(1) = 0$ | $\xi^4 - 1.5\xi^3 + 0.5\xi$ |

**Table 5.** Simple-simple fundamental mode function ($\lambda_1$).

| Boundary Condition | Mathematical Expression | Fundamental Mode Function |
|---|---|---|
| GBC (2) + NBC (2) | $\lambda_1(0) = 0,\ \lambda_1(1) = 0$ <br> $\lambda_1''(0) = 0,\ \lambda_1''(1) = 0$ | $\xi^4 - 2\xi^3 + \xi$ |

The expansion to the higher-order term for the fundamental mode function for each boundary condition defined in Tables 3–5 is the same as the method mentioned in Equations (11) to (16). It shows the mass and stiffness matrix for the vertical member among the three component structures with the expanded mode function, as shown in Equations (31) and (32):

$$[M_A] = mL_A \begin{vmatrix} \psi_1\psi_1 & \cdots & \psi_1\psi_m & \psi_1\phi_1 & \cdots & \psi_1\phi_m & \psi_1\gamma_1 & \cdots & \psi_1\gamma_m \\ \vdots & \ddots & \vdots & \vdots & \ddots & \vdots & \vdots & \ddots & \vdots \\ \psi_m\psi_1 & \cdots & \psi_m\psi_m & \psi_m\phi_1 & \cdots & \psi_m\phi_m & \psi_m\gamma_1 & \cdots & \psi_m\gamma_m \\ \phi_1\psi_1 & \cdots & \phi_1\psi_m & \phi_1\phi_1 & \cdots & \phi_1\phi_m & \phi_1\gamma_1 & \cdots & \phi_1\gamma_m \\ \vdots & \ddots & \vdots & \vdots & \ddots & \vdots & \vdots & \ddots & \vdots \\ \phi_m\psi_1 & \cdots & \phi_m\psi_m & \phi_m\phi_1 & \cdots & \phi_m\phi_m & \phi_m\gamma_1 & \cdots & \phi_m\gamma_m \\ \gamma_1\psi_1 & \cdots & \gamma_1\psi_m & \gamma_1\phi_1 & \cdots & \gamma_1\phi_m & \gamma_1\gamma_1 & \cdots & \gamma_1\gamma_m \\ \vdots & \ddots & \vdots & \vdots & \ddots & \vdots & \vdots & \ddots & \vdots \\ \gamma_m\psi_1 & \cdots & \gamma_m\psi_m & \gamma_m\phi_1 & \cdots & \gamma_m\phi_m & \gamma_m\gamma_1 & \cdots & \gamma_m\gamma_m \end{vmatrix} \quad (31)$$

$$[K_A] = \frac{8EI}{L_A^3} \begin{vmatrix} \psi_1''\psi_1'' & \cdots & \psi_1''\psi_m'' & \psi_1''\phi_1'' & \cdots & \psi_1''\phi_m'' & \psi_1''\gamma_1'' & \cdots & \psi_1''\gamma_m'' \\ \vdots & \ddots & \vdots & \vdots & \ddots & \vdots & \vdots & \ddots & \vdots \\ \psi_m''\psi_1'' & \cdots & \psi_m''\psi_m'' & \psi_m''\phi_1'' & \cdots & \psi_m''\phi_m'' & \psi_m''\gamma_1'' & \cdots & \psi_m''\gamma_m'' \\ \phi_1''\psi_1'' & \cdots & \phi_1''\psi_m'' & \phi_1''\phi_1'' & \cdots & \phi_1''\phi_m'' & \phi_1''\gamma_1'' & \cdots & \phi_1''\gamma_m'' \\ \vdots & \ddots & \vdots & \vdots & \ddots & \vdots & \vdots & \ddots & \vdots \\ \phi_m''\psi_1'' & \cdots & \phi_m''\psi_m'' & \phi_m''\phi_1'' & \cdots & \phi_m''\phi_m'' & \phi_m''\gamma_1'' & \cdots & \phi_m''\gamma_m'' \\ \gamma_1''\psi_1'' & \cdots & \gamma_1''\psi_m'' & \gamma_1''\phi_1'' & \cdots & \gamma_1''\phi_m'' & \gamma_1''\gamma_1'' & \cdots & \gamma_1''\gamma_m'' \\ \vdots & \ddots & \vdots & \vdots & \ddots & \vdots & \vdots & \ddots & \vdots \\ \gamma_m''\psi_1'' & \cdots & \gamma_m''\psi_m'' & \gamma_m''\phi_1'' & \cdots & \gamma_m''\phi_m'' & \gamma_m''\gamma_1'' & \cdots & \gamma_m''\gamma_m'' \end{vmatrix} \quad (32)$$

where we can express the mass matrix as Equation (33) by using orthogonality:

$$[M_A] = mL_A \begin{vmatrix} 1 & \cdots & 0 & \psi_1\phi_1 & \cdots & \psi_1\phi_m & \psi_1\gamma_1 & \cdots & \psi_1\gamma_m \\ \vdots & \ddots & \vdots & \vdots & \ddots & \vdots & \vdots & \ddots & \vdots \\ 0 & \cdots & 1 & \psi_m\phi_1 & \cdots & \psi_m\phi_m & \psi_m\gamma_1 & \cdots & \psi_m\gamma_m \\ \phi_1\psi_1 & \cdots & \phi_1\psi_m & 1 & \cdots & 0 & \phi_1\gamma_1 & \cdots & \phi_1\gamma_m \\ \vdots & \ddots & \vdots & \vdots & \ddots & \vdots & \vdots & \ddots & \vdots \\ \phi_m\psi_1 & \cdots & \phi_m\psi_m & 0 & \cdots & 1 & \phi_m\gamma_1 & \cdots & \phi_m\gamma_m \\ \gamma_1\psi_1 & \cdots & \gamma_1\psi_m & \gamma_1\phi_1 & \cdots & \gamma_1\phi_m & 1 & \cdots & 0 \\ \vdots & \ddots & \vdots & \vdots & \ddots & \vdots & \vdots & \ddots & \vdots \\ \gamma_m\psi_1 & \cdots & \gamma_m\psi_m & \gamma_m\phi_1 & \cdots & \gamma_m\phi_m & 0 & \cdots & 1 \end{vmatrix} \tag{33}$$

and $M_B$, $M_C$, and $K_B$, $K_C$ can be expressed similarly to Equations (31) and (32).

Therefore, the mass and stiffness matrix of the subcomponents in Figure 2 can be expressed as in Equations (34) and (35).

$$[M] = \begin{vmatrix} M_A & 0 & 0 \\ 0 & M_B & 0 \\ 0 & 0 & M_C \end{vmatrix} \tag{34}$$

$$[K] = \begin{vmatrix} K_A & 0 & 0 \\ 0 & K_B & 0 \\ 0 & 0 & K_C \end{vmatrix} \tag{35}$$

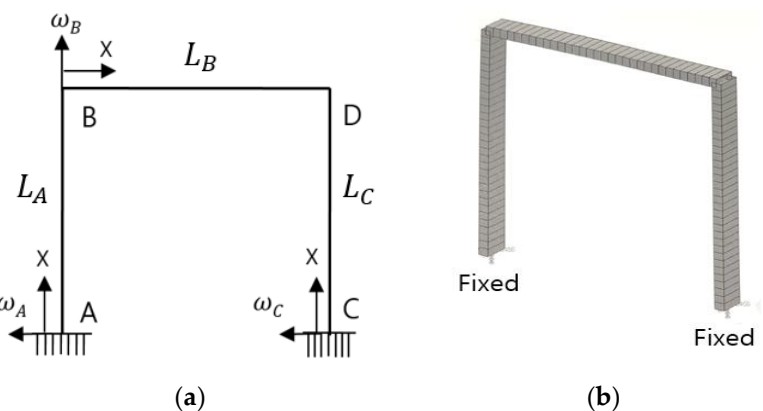

(**a**)                                                    (**b**)

**Figure 2.** Structure model of three components type connected beam, (**a**) simplified model (**b**) FEA model.

Note that the number of generalized coordinates shall be 3 ($m + n + k$), since the number of constraints at junction $B$, $D$ are 2, separately.

Displacement continuity:

$$w_A(1) - u_B(0) = 0 \tag{36}$$

$$w_C(1) - u_B(1) = 0 \tag{37}$$

Slope continuity:

$$w_A{}'(1) - w_B{}'(0) = 0 \tag{38}$$

$$w_A{}'(1) - w_B{}'(1) = 0 \tag{39}$$

The coordinate system reflecting the constraints is expressed in Equation (40):

$$
\left\{
\begin{array}{c}
p_{A1} \\
\vdots \\
p_{Am} \\
q_{A1} \\
\vdots \\
q_{Am} \\
r_{A1} \\
\vdots \\
r_{Am} \\
q_{B1} \\
\vdots \\
q_{Bm} \\
S_{B1} \\
\vdots \\
S_{Bn} \\
O_{B1} \\
\vdots \\
O_{Bn} \\
p_{C1} \\
\vdots \\
p_{Ck} \\
q_{C1} \\
\vdots \\
q_{Ck} \\
r_{C1} \\
\vdots \\
r_{Ck} \\
u_{b0} \\
u_{b1}
\end{array}
\right\}
= [C]
\left\{
\begin{array}{c}
p_{A1} \\
\vdots \\
p_{Am} \\
q_{A1} \\
\vdots \\
q_{Am} \\
r_{A1} \\
\vdots \\
r_{Am} \\
q_{B1} \\
\vdots \\
q_{Bn-1} \\
S_{B1} \\
\vdots \\
S_{Bn-1} \\
O_{B1} \\
\vdots \\
O_{Bn} \\
p_{C1} \\
\vdots \\
p_{Ck} \\
q_{C1} \\
\vdots \\
q_{Ck} \\
r_{C1} \\
\vdots \\
r_{Ck}
\end{array}
\right\}
\tag{40}
$$

where the [C] matrix represents a constraint and is implemented in the same way as Equation (18).

The [M], [K], and [C] matrices implemented in this way were substituted into the Lagrange equation of motion to calculate the natural frequencies, and the results are mentioned in Section 4.

## 3. Proposal of Methodology for Simplification of Computation

### 3.1. Two Components Type Connected Structure

Frequently, we may more concern about the vibration of one subcomponent. Suppose we concern the vibration of part OA in Figure 1.

The suitable boundary conditions at junction O can be obtained by removal of generalized coordinates of subcomponent OB through satisfaction of constraints at junction O as shown in Equations (41) and (42):

$$q_{Bj} = \alpha q_{Ai} \tag{41}$$

$$p_{Bj} = -\alpha^2 p_{Ai} \text{ for } i = 1 \text{ to } m \ (m = n) \tag{42}$$

where $\alpha$ is the ratio of length for subcomponents ($\alpha = L_B/L_A$):

$$W_A(\zeta, t) = \sum_{i=1}^{m} (\psi_i(\zeta)p_{Ai}(t) + \phi_i(\zeta)q_{Ai}(t)) \tag{43}$$

$$W_B(\xi, t) = \sum_{j=1}^{n} \left(\psi_j(\xi) \times -\alpha^2 p_{Ai}(t) + \phi_j(\xi) \times \alpha q_{Ai}(t)\right) \tag{44}$$

Although the assumption of simplification by the constraint at the junction O is excessive, the natural frequency and mode shape in the range $\alpha = 0$ to $1$ are similar to the FEA results. Therefore, we may expect that this will give reasonable result for all cases ($0 \leq \alpha \leq 1$).

$$[M] = mL_A \begin{vmatrix} \psi_1\psi_1(1+\alpha^5) & \cdots & \psi_1\psi_n(1+\alpha^5) & \psi_1\phi_1(1-\alpha^4) & \cdots & \psi_1\phi_n(1-\alpha^4) \\ \vdots & \ddots & \vdots & \vdots & & \vdots \\ \psi_m\psi_1(1+\alpha^5) & \cdots & \psi_m\psi_n(1+\alpha^5) & \psi_m\phi_1(1-\alpha^4) & \cdots & \psi_m\phi_n(1-\alpha^4) \\ \phi_1\psi_1(1-\alpha^4) & \cdots & \phi_1\psi_n(1-\alpha^4) & \phi_1\phi_1(1+\alpha^3) & \cdots & \phi_1\phi_n(1+\alpha^3) \\ \vdots & \ddots & \vdots & \vdots & & \vdots \\ \phi_m\psi_1(1-\alpha^4) & \cdots & \phi_m\psi_n(1-\alpha^4) & \phi_m\phi_1(1+\alpha^3) & \cdots & \phi_m\phi_n(1+\alpha^3) \end{vmatrix} \tag{45}$$

$$[K] = \frac{16EI}{L_A^3} \begin{vmatrix} \psi_1''\psi_1''(1+\alpha) & \cdots & \psi_1''\psi_n''(1+\alpha) & 0 & \cdots & 0 \\ \vdots & \ddots & \vdots & \vdots & & \vdots \\ \psi_m''\psi_1''(1+\alpha) & \cdots & \psi_m''\psi_n''(1+\alpha) & 0 & \cdots & 0 \\ 0 & \cdots & 0 & \phi_1''\phi_1''\left(1+\frac{1}{\alpha}\right) & \cdots & \phi_1''\phi_n''\left(1+\frac{1}{\alpha}\right) \\ \vdots & \ddots & \vdots & \vdots & & \vdots \\ 0 & \cdots & 0 & \phi_m''\phi_1''\left(1+\frac{1}{\alpha}\right) & \cdots & \phi_m''\phi_n''\left(1+\frac{1}{\alpha}\right) \end{vmatrix} \tag{46}$$

The mass and stiffness matrix can be created from Equations (43) and (44). Then, the degree of freedom can reduce 50% compared to the previous calculation method, and mass and stiffness matrix are shown in Equations (45) and (46).

The [M] and [K] matrices were substituted into the Lagrange equation of motion to calculate the natural frequencies, and the results are mentioned in Section 4.

### 3.2. Three Components Type Connected Structure

In the case of a structures in Figure 2, these structures have the symmetric and asymmetric modes as like Figure 3. In order to reflect the behavior of the structure and simplify the calculation, the mode function was applied separately according to the symmetric and asymmetric modes.

Firstly, in the case of asymmetry, the structure has a mode shape similar to the simple support condition affected by the slope at the middle point of the horizontal member, and in the case of symmetry, it has a mode similar to the fixed support with the slope close to 0 at the middle point.

Of course, it is not strictly a fixed condition because deflection occurs at the middle point, but it is suitable as a simplification method as it satisfies the allowable range of analysis when calculated considering the fixed boundary condition.

This can be seen visually in the FEA results.

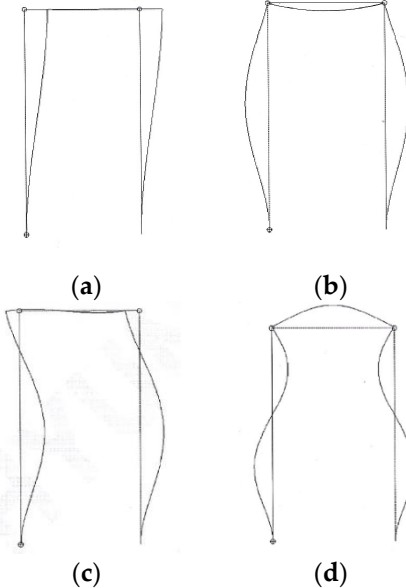

**Figure 3.** FEA Result of n-type structure: (**a**) 1st mode—asymmetry; (**b**) 2nd mode—symmetry; (**c**) 3rd mode—asymmetry; (**d**) 4th mode—symmetry.

In the case of three components type structure, Figures 4 and 5 show the shape of the symmetric and asymmetric modes in the intermediate position. It can be concluded that this can be implemented by a combination of mode functions suitable for conditions in symmetrical and asymmetry.

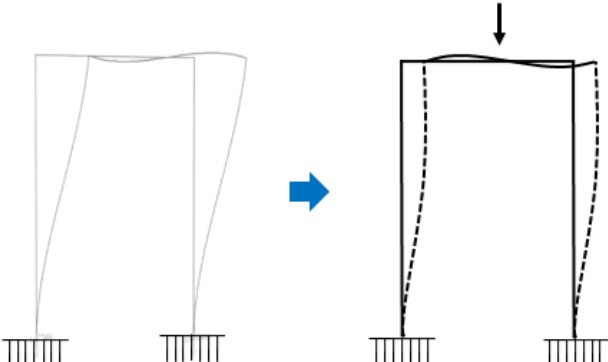

**Figure 4.** Idealization of approximate analytical approach for asymmetric mode.

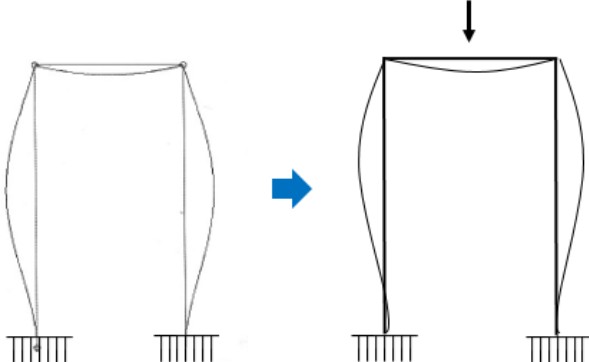

**Figure 5.** Idealization of approximate analytical approach for symmetric mode.

In the case of asymmetry as shown in Figure 4, the mode function of a simple boundary

condition of 1/2 L (L: the length of horizontal beam) length is used for the horizontal member to the mode function of the existing vertical member, and in the case of Figure 5, the mode function of the fixed boundary condition is used to compare the results. The results performed in Section 4.2 were shown valid results within 15%.

In Table 6, the mode shapes for length ratios of 5 m:2.5 m:5 m are shown, the mode of 5 m:5 m:5 m is similar to the previous case.

**Table 6.** The mode shape for each length ratio ($L_A$:$L_B$:$Lc$ = 5 m:2.5 m:5 m).

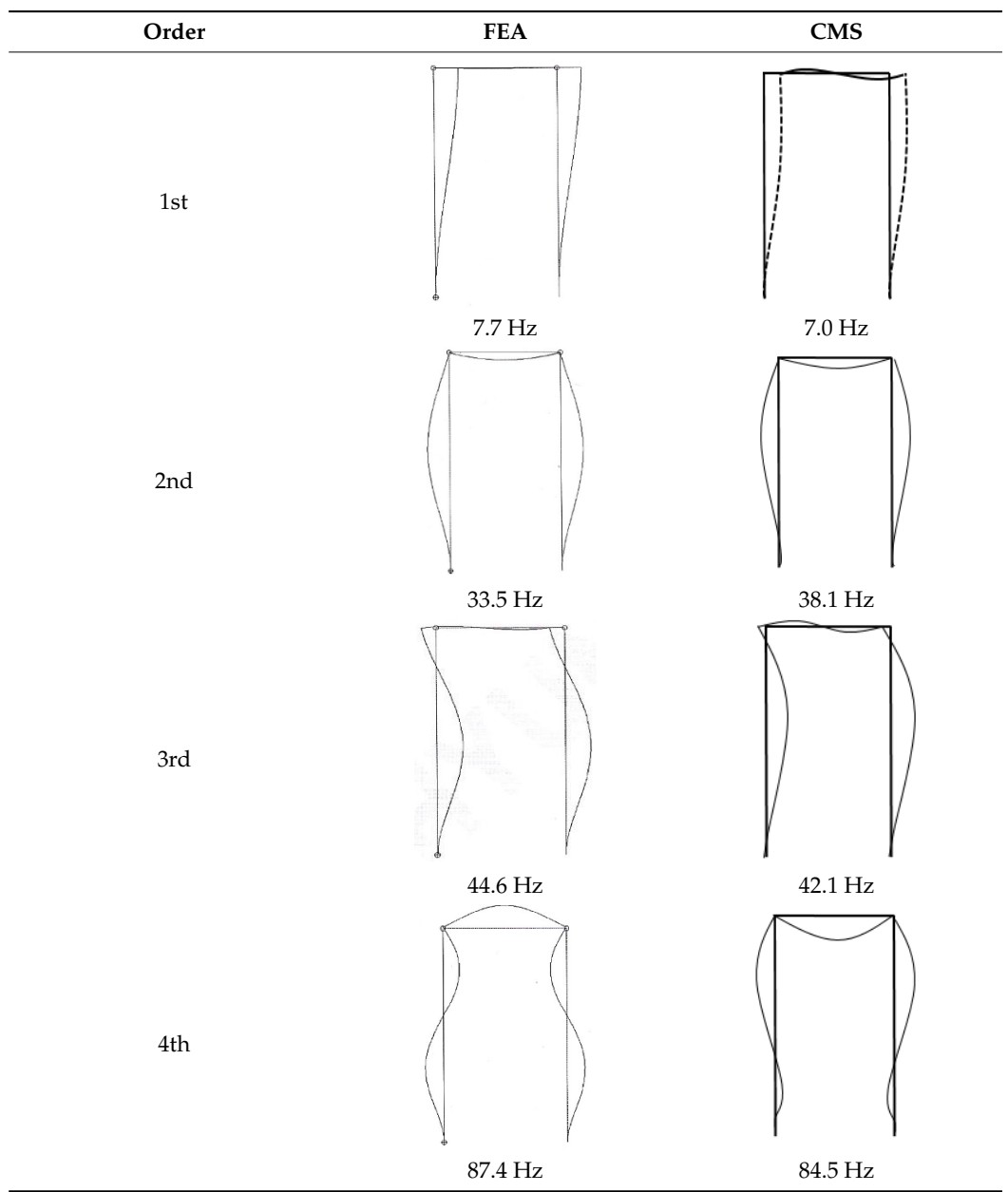

| Order | FEA | CMS |
|:-----:|:---:|:---:|
| 1st | 7.7 Hz | 7.0 Hz |
| 2nd | 33.5 Hz | 38.1 Hz |
| 3rd | 44.6 Hz | 42.1 Hz |
| 4th | 87.4 Hz | 84.5 Hz |

## 4. Comparison of Numerical Results and Mode Shape

*4.1. Comparison of Numerical Calculation Results by Fundamental Mode Function*

The difference was confirmed by applying the fundamental mode function for each boundary condition defined above to the two components type connected structure.

The application cases are described in Table 7, and the results are shown in Table 8.

**Table 7.** The case of study for two components type.

| Case | $W_A$ | $W_B$ | Junction Constraint |
|---|---|---|---|
| Case 1 | F-F (4) + F-S (3) | F-F (4) + F-S (3) | |
| Case 2 | | F-F (4) + F-S (4) | Slope and Moment |
| Case 3 | F-F (4) + F-S (4) | F-F (4) + F-S (3) | |
| Case 4 | | F-F (4) + F-S (4) | |
| Case 5 | F-F (4) + F-S (3) | F-F (4) + F-S (3) | |
| Case 6 | | F-F (4) + F-S (4) | Slope Only |
| Case 7 | F-F (4) + F-S (4) | F-F (4) + F-S (3) | |
| Case 8 | | F-F (4) + F-S (4) | |

**Table 8.** The result of numerical calculation for fundamental mode of two components type.

| $L_A$:$L_B$ | Order | FEA | Case 1 | Case 2 | Case 3 | Case 4 | Case 5 | Case 6 | Case 7 | Case 8 |
|---|---|---|---|---|---|---|---|---|---|---|
| | 1 | 29.1 | 29.4 | 29.4 | 29.4 | 29.4 | 29.4 | 29.4 | 29.4 | 29.4 |
| 1:1 | 2 | 41.9 | 42.6 | 42.6 | 42.6 | 42.6 | 42.6 | 42.6 | 42.6 | 42.6 |
| | 3 | 92.9 | 95.3 | 95.3 | 95.3 | 95.3 | 95.3 | 95.3 | 95.3 | 95.3 |
| | 1 | 35.2 | 39.2 | 39.2 | 39.2 | 39.2 | 39.1 | 39.1 | 39.1 | 39.1 |
| 1:0.6 | 2 | 86.5 | 85.8 | 85.8 | 85.8 | 85.8 | 85.6 | 85.6 | 85.6 | 85.6 |
| | 3 | 112.7 | 119.6 | 119.6 | 119.6 | 119.6 | 119.5 | 119.5 | 119.5 | 119.5 |
| | 1 | 36.8 | 42.6 | 42.6 | 42.6 | 42.6 | 42.5 | 42.5 | 42.5 | 42.5 |
| 1:0.4 | 2 | 101.8 | 116.1 | 116.1 | 116.1 | 116.1 | 116.0 | 116.0 | 116.0 | 116.0 |
| | 3 | 180.6 | 187.3 | 187.3 | 187.3 | 187.3 | 187.3 | 187.3 | 187.3 | 187.3 |

From the result of above study, it can be seen that the contribution of the geometric boundary condition to the natural frequency is dominant than the natural boundary condition. In addition, it was confirmed that the influence was insignificant even when moment continuity was considered as a constraint condition at the junction.

*4.2. Two Components Type Connected Structure*

The properties of structures used in the numerical analysis are shown in Table 1. Table 9 shows the results of component mode synthesis using the proposed polynomial function and the numerical results using the proposed simplification method.

**Table 9.** Comparison of numerical result (Unit:Hz).

| Ratio ($L_A$:$L_B$) | | 1st | 2nd | 3rd | Ratio ($L_A$:$L_B$) | | 1st | 2nd | 3rd |
|---|---|---|---|---|---|---|---|---|---|
| | FEA | 29.1 | 41.9 | 92.9 | | FEA | 36.0 | 97.8 | 136.8 |
| 1:1 | CMS | 29.1 | 42.5 | 95.2 | 1:0.5 | CMS | 36.3 | 100.7 | 146.8 |
| | PMSC | 29.3 | 42.5 | 98.8 | | PMSC | 36.4 | 101.6 | 147.5 |
| | FEA | 33.2 | 56.5 | 102.6 | | FEA | 36.8 | 101.8 | 180.6 |
| 1:0.8 | CMS | 33.3 | 57.6 | 106.2 | 1:0.4 | CMS | 37.1 | 105.4 | 197.3 |
| | PMSC | 33.5 | 57.7 | 108.9 | | PMSC | 37.2 | 106.3 | 200.9 |
| | FEA | 35.2 | 86.5 | 112.7 | | FEA | 38.4 | 105.9 | 205.6 |
| 1:0.6 | CMS | 35.4 | 88.7 | 118.8 | 1:0.2 | CMS | 39.2 | 110.9 | 224.8 |
| | PMSC | 35.6 | 89.2 | 120.1 | | PMSC | 39.3 | 111.5 | 233.8 |

CMS: Component mode synthesis. PMSC: Proposal of methodology for simplification of computation.

In Tables 10 and 11, the mode shapes for length ratios of 1:1, 1:0.5 are shown, respectively.

**Table 10.** The mode shape for each length ratio ($L_A$:$L_B$ = 1:1).

| Order | FEA | CMS | PMSC |
|---|---|---|---|
| 1st | 29.1 Hz | 29.1 Hz | 29.3 Hz |
| 2nd | 41.9 Hz | 42.5 Hz | 42.5 Hz |
| 3rd | 92.9 Hz | 95.2 Hz | 98.8 Hz |

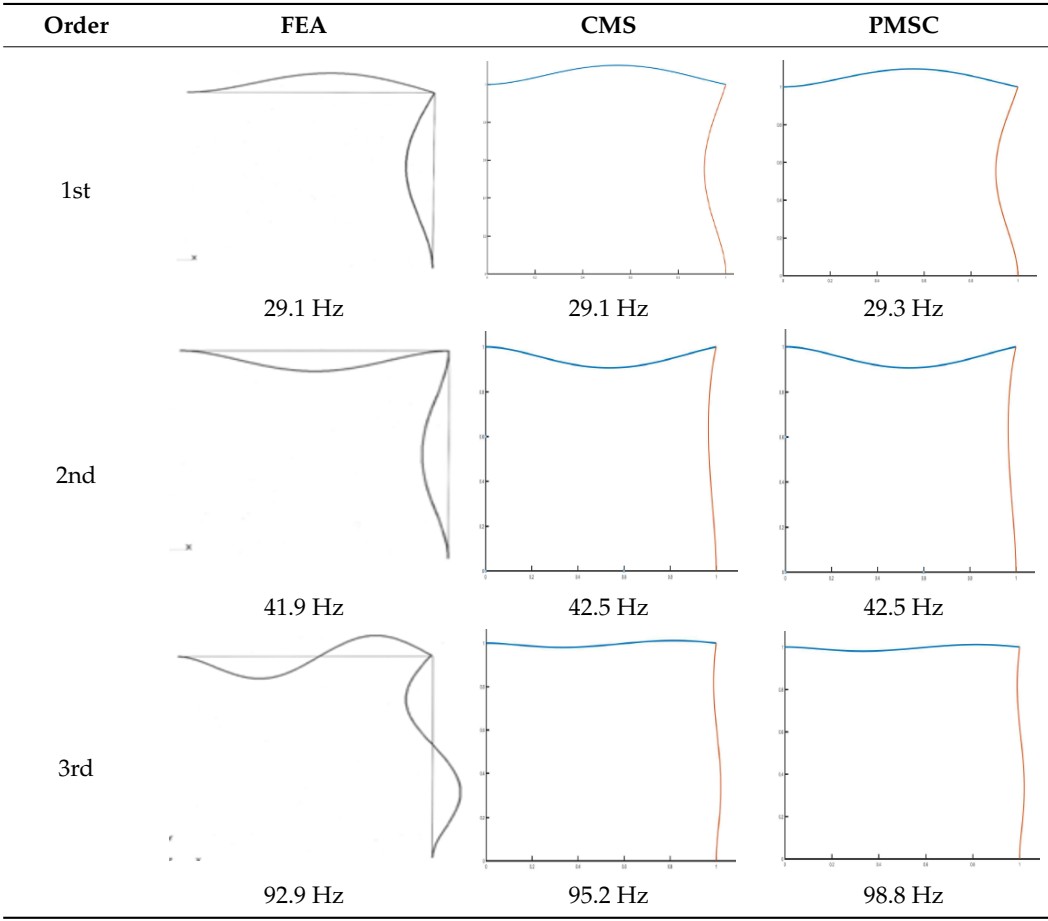

**Table 11.** The mode shape for each length ratio ($L_A$:$L_B$ = 1:0.5).

| Order | FEA | CMS | PMSC |
|---|---|---|---|
| 1st | 36.0 Hz | 36.3 Hz | 36.4 Hz |
| 2nd | 97.8 Hz | 100.7 Hz | 101.6 Hz |
| 3rd | 136.8 Hz | 146.8 Hz | 147.5 Hz |

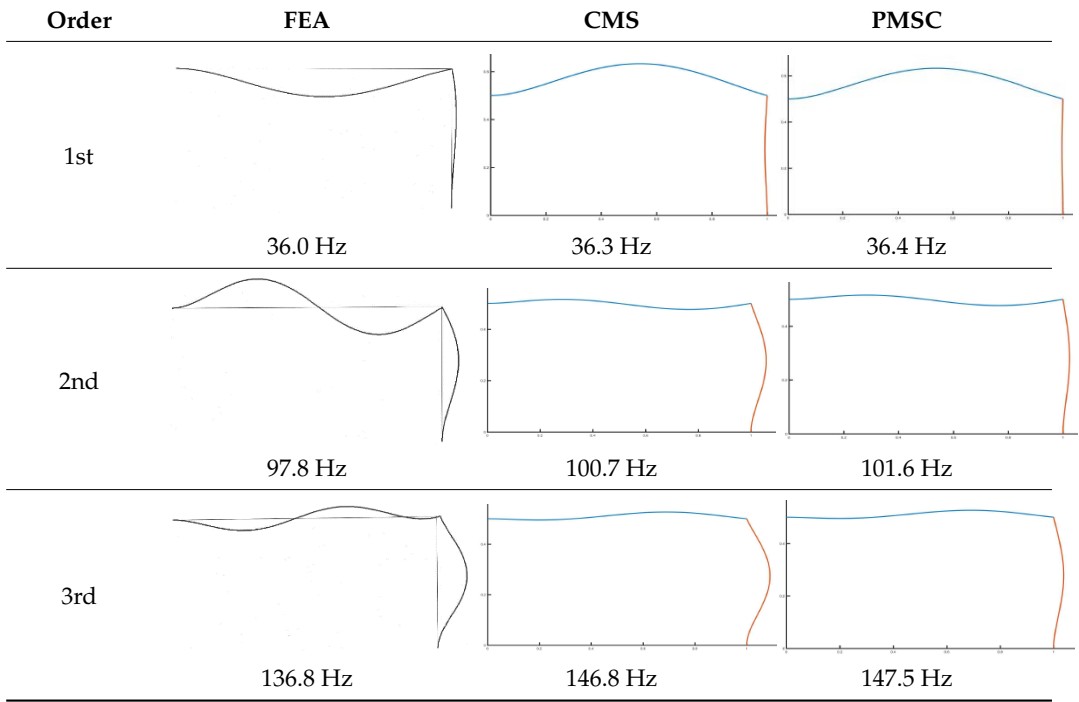

*4.3. Three Components Type Connected Structure*

The properties of structures used in the numerical analysis are shown in Table 2. Table 12 shows the results of component mode synthesis using the proposed polynomial function and the numerical results using the proposed simplification method, and the mode shape is shown in Section 3.2.

**Table 12.** Comparison of numerical result (Unit:Hz).

| Ratio ($L_A$:$L_B$:$L_C$) | Calculation Method | 1st | 2nd | 3rd | 4th |
|---|---|---|---|---|---|
| 5 m:5 m:5 m | FEA | 6.1 | 23.9 | 38.9 | 41.9 |
| | CMS | 6.7 | 20.8 | 37.4 | 42.0 |
| | PMSC | 6.8 | 21.2 | 37.7 | 42.4 |
| 5 m:2.5 m:5 m | FEA | 7.7 | 33.5 | 44.6 | 87.4 |
| | CMS | 7.0 | 38.1 | 42.1 | 84.5 |
| | PMSC | 7.3 | 41.2 | 44.3 | 86.2 |

## 5. Conclusions

We proposed the use of polynomials that can satisfy the boundary conditions at the junction between subcomponents, and a method that can be calculated by dramatically reducing the infinitely increasing degree of freedom.

To do this, numerical results of a structural subcomponent OA considering dynamic and static coupling of subcomponent OB are also given, and these numerical results are also compared with result of FEA. Numerical results prove the following:

1. In the two and three components, which are the typical shapes of the structure, it is proposed that the effective boundary conditions of the junction can be implemented by a combination of the appropriate boundary conditions of fixed, simple, and free.
2. Proposed polynomial mode functions allows wide and mode effective application of component mode synthesis for the vibration analysis of many ship local structures.
3. Numerical results based on the suggested method to reflect the dynamic and static coupling of connected subcomponent are proved to show very good agreement.
4. The proposed polynomial function is efficient enough to be compared with the FEA results in terms of natural frequency and mode.
5. The application of method suggested can be easily expanded for the analysis of Timoshenko beam or other more complicated structures such as reinforced plates.

**Author Contributions:** Conceptualization, J.H.P., D.Y.Y.; methodology, J.H.P.; D.Y.Y.; software, J.H.P.; validation, J.H.P., D.Y.Y.; formal analysis, J.H.P., D.Y.Y.; investigation, J.H.P.; writing—original draft preparation, J.H.P., D.Y.Y.; writing—review and editing, J.H.P., D.Y.Y.; funding acquisition, J.H.P. All authors have read and agreed to the published version of the manuscript.

**Funding:** This research received no external funding.

**Conflicts of Interest:** The authors declare no conflict of interest.

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
