# Peer review of "A Proposal of Mode Polynomials for Efficient Use of Component Mode Synthesis and Methodology to Simplify the Calculation of the Connecting Beams"

_jmse, doi:10.3390/jmse9010020_

Round 1
Reviewer 1 Report
This paper suggested polynomials for simple and fixed supports to represent subsystems in ships. In general, there are two assumptions that the authors should provide more descriptions. i) The authors used the Euler beam assumption. Therefore, they should provide the substructure dimension (e.g., Figure 1a) to show the substructure's slenderness. If the slenderness is too small, they should use the Timoshenko beam model instead. ii) The authors assume that the connection is either fixed or simple support. In reality, the connection should be a combination of both fixed and simple supports. The authors could simplify the connection as a linear combination of both supports, which will better describe the substructure's dynamic characteristics. In general, the English could be improved. Detailed comments:
- In the Abstract, when first mention the term 'FEA,' the authors should give the full name.
- The Abstract lacks a summarization of the conclusion.
- Line 37, what is the damage type in Figure 1a?
- Line 97, the definition of P_Ai(t), P_Bi(t), x, t, and m should be shown following equations (1) and (2).
- Line 108, the authors should explain the reasons that they assume fourth-order polynomial and boundary conditions.
- Line 109, Line 171, in Equations (5) and (27), the 1st order polynomial is missed.
- Line 137, why there is no coupling between OA and OB?
- Line 147, at the beginning of section 2.2, the authors should introduce the function of the substruction in Figure3a. Also, in Figure 3b, the two columns in the substructure seem to have an angle. Shouldn't this angle be considered in the simplified model (Figure 3c)?
- Line 154, because loadings from both directions should be considered, the substructure is symmetrical. Therefore, the authors can consider only one column and half beam, precisely the same as the model shown in Figure 2. Section 2.1 and 2.2 can be combined.
- Line 90 and 206, the titles of Sections 2.1 and 3.1 are not complete. What type?
- Line 201, the C matrix should be written as [C], not |C|.
- Mode shapes should be given in Section 3.1, similar to Figure 4.
- There are no line numbers since Section 3.2.

Author Response
i) The authors used the Euler beam assumption. Therefore, they should provide the substructure dimension (e.g., Figure 1a) to show the substructure's slenderness. If the slenderness is too small, they should use the Timoshenko beam model instead.
A) The cross-sectional shape of the structural model and the length of the model are shown in the manuscript.
ii) The authors assume that the connection is either fixed or simple support. In reality, the connection should be a combination of both fixed and simple supports. The authors could simplify the connection as a linear combination of both supports, which will better describe the substructure's dynamic characteristics. In general, the English could be improved.
A) The author has the same idea as the reviewer. This was mentioned in lines 91~93 of the original manuscript. In the revised manuscript, this is mentioned with more emphasis.
I have written more carefully about the English expression.
1. In the Abstract, when first mention the term 'FEA,' the authors should give the full name.
A) We have modified the manuscript for the pointed out.
2. The Abstract lacks a summarization of the conclusion.
A) A summary of the manuscript is included in the abstract section.
3. Line 37, what is the damage type in Figure 1a?
A) The type of damage to the tank wall is Crack, which is mentioned in the manuscript.
4. Line 97, the definition of P_Ai(t), P_Bi(t), x, t, and m should be shown following equations (1) and (2).
A) We entered after checking all the missing subscripts.
5. Line 108, the authors should explain the reasons that they assume fourth-order polynomial and boundary conditions.
A) In this paper, a polynomial that reflects Euler beam property was used, and since the Euler beam equation obtains a general solution from the fourth order differential equation, the fourth order equation was used as the fundamental mode function. In the revised manuscript, geometric boundary conditions and natural boundary conditions are classified for each boundary condition. The mode function is defined by dividing the third and fourth order functions, and the results are compared.
6. Line 109, Line 171, in Equations (5) and (27), the 1st order polynomial is missed.
A) As a mode function defined using Euler beam property, the fourth order equation function has been defined as first order mode function. And, in the revised paper, the case where the first order mode function starts with a third equation and a fourth equation according to the boundary conditions is described, and the results are compared.
7. Line 137, why there is no coupling between OA and OB?
A) The defined functions between the horizontal and vertical members are fixed-fixed and fixed-simple functions, and the displacement at the end is zero, so it is used to mean that there is no influence on the displacement between them.
8. Line 147, at the beginning of section 2.2, the authors should introduce the function of the substruction in Figure3a. Also, in Figure 3b, the two columns in the substructure seem to have an angle. Shouldn't this angle be considered in the simplified model (Figure 3c)?
A) Figures 3a and 3b were deleted from the revised manuscript, as it was determined that it was not related to the understanding of the manuscript. For reference, we have added answers to inquiries.
The function of Figure 3a is a truss platform for access such as operation of the upper valve of the LNG fuel tank installed at the stern of a 180k bulk carrier vessel.
And, although the structure has an angle, it is assumed that the slope continuity is maintained as it is actually supported by a bracket structure between the two structures.
9. Line 154, because loadings from both directions should be considered, the substructure is symmetrical. Therefore, the authors can consider only one column and half beam, precisely the same as the model shown in Figure 2. Section 2.1 and 2.2 can be combined.
A) The author has the same idea as the reviewer. In the case of a actual structure, it is extremely rare that the structure movement occurs due to a both directions load, and it is judged to be at a level that can be ignored even if deformation occurs.
Therefore, I think the reviewer's suggestion that it can be combined in symmetrical mode is correct. However, in this study, two methods were mentioned in order to present a solution that can be solved even in the case of asymmetric cases, and a method for this was suggested.
10. Line 90 and 206, the titles of Sections 2.1 and 3.1 are not complete. What type?
A) It seems that the original manuscript was not in English, so it was deleted during the editing process. This is expressed in English in the revised manuscript.
11. Line 201, the C matrix should be written as [C], not |C|.
A) Corrected the expression.
12. Mode shapes should be given in Section 3.1, similar to Figure 4.
A) We understood that the mode shape should be placed in similar section, and the arrangement was corrected.
13. There are no line numbers since Section 3.2.
A) Added line number
Reviewer 2 Report
The present paper shows an analytical approach for the evaluation of the modal properties of structures made by connected beams. The Rayleigh-Ritz method is used, and the present method provides a modal shapes estimation which can be useful as a starting point for the design of such structures. The paper reports numerical results and proves its validity through comparison with FEA results, and the publication is suggested after minor revision are made:
1) In the introduction part, it is worthy to mention the following paper:
Carrera, E., et al. "Nonlinear analysis of thin-walled beams with highly deformable sections." International Journal of Non-Linear Mechanics 128: 103613.
Although it provides a geometrical nonlinear approach, it reports a novel approach for the FE analysis of connected-beams structure, which is the main topic of the author's work.
In addition, if the geometrical nonlinear topic wants to be mentioned (and i think it is needed to do), the following paper could be mentioned as vibration analysis in the geom NL field:
Pagani, A., R. Augello, and E. Carrera. "Frequency and mode change in the large deflection and post-buckling of compact and thin-walled beams." Journal of Sound and Vibration 432 (2018): 88-104.
Carrera, E., A. Pagani, and R. Augello. "Effect of large displacements on the linearized vibration of composite beams." International Journal of Non-Linear Mechanics 120 (2020): 103390.
2) The FEA results are shown, but no evidence of the type of which FE is addressed. Could the authors provide it? In terms of DOF, connection and type of elements.
3) The modal shapes are evaluated and the matching between CMS and FEA results are shown. By the way, the paper lacks the information about the natural frequency. Why the authors omitted such important information? It should be provided.
4) Tables 3 and 4 could be changed: the authors should show the same scale factor for the modal shapes for a better comparison
5) An overall improvement on the image quality could be achieved
Author Response
1) In the introduction part, it is worthy to mention the following paper:
A) The papers recommended by the reviewers were reviewed and added to the introduction and references.
2) The FEA results are shown, but no evidence of the type of which FE is addressed. Could the authors provide it? In terms of DOF, connection and type of elements.
A) The property of elements such as the structural shape and length used for the FEM model and calculation were entered into a table.
3) The modal shapes are evaluated and the matching between CMS and FEA results are shown. By the way, the paper lacks the information about the natural frequency. Why the authors omitted such important information? It should be provided.
A) The natural frequency is displayed on each mode shape.
4) Tables 3 and 4 could be changed: the authors should show the same scale factor for the modal shapes for a better comparison
A) Adjusted the scale factor of the figure and table.
5) An overall improvement on the image quality could be achieved
A) I tried to make the picture more clearness, but due to personal circumstances, some of the pictures are printed as hardcopy, so the sharpness seems to decrease during the capture process.
I tried to improve this problem as much as possible, but I am sorry that it has not been done as much as I wanted.
Reviewer 3 Report
The paper proposes a method aiming at analyzing dynamic behavior of single structural elements by means of analytical approaches and suitable boundary conditions accounting for the interaction with the rest of the structure. The document has to be rejected since an attempt of auto-plagiarism has been detected. Furthermore, any significant novelty can be inferred and the document is poorly written, as detailed in the following.
1. The first part of the introduction is identical to the beginning of the paper [1], already published by the authors in another journal. In a similar manner, Figure 1 of the submitted document is identical to Figure 1 of the paper mentioned above. The citation has not been properly referenced and the paper [1] is not even in the reference list. This shapes up to be an attempt of plagiarism.
2. Any substantial novelty can be inferred from the document. Analytical methods are well-known and documented in the literature. Also the replacement of the structural element with equivalent constraint seems not new, but highly reminiscent of the direct stiffness method, a basic approach for the structural mechanics, use also before the advent of computers.
3. The document is poorly written. In the following a non-comprehensive list of detected problems is reported.
i. Improper paragraph subdivision of the document. Why most of paragraphs are just one sentence long?
ii. geometrical conditions and natural conditions at junctions (Line 13) repeated word conditions
iii. Polynomials (Line 14) wrong use of capital letter
iv. each shipyard has developed and used an approximate vibration calculation program according to the circumstances of the design. (Line 29) vague statement, it is more reasonable to believe that practitioners use commercial software nowadays. Provide suitable references.
v. the beam function is used in the assumed mode function (Line 42) what is the meaning of this sentence?
vi. This method is an introduction to a methodology (Line 72) tautology.
vii. The words coordinate and variable have different meaning. Conversely authors use them as synonymous e.g., if wa is a coordinate (Figure 2), it can not be also the displacement variable (Equations (1) and (2)).
viii. Figure 6, wrong representation of the symmetry constraints.
ix. Section 2.2 looks like the copy paste of Section 2.1. It just consider 3 segments instead of 2.
References
[1] Park, J.-H. and J.-H. Yang (2020). Normal mode analysis for connected plate structure using efficient mode polynomials with component mode synthesis. Applied Sciences 10(21), 7717.
Author Response
1. The first part of the introduction is identical to the beginning of the paper [1], already published by the authors in another journal. In a similar manner, Figure 1 of the submitted document is identical to Figure 1 of the paper mentioned above. The citation has not been properly referenced and the paper [1] is not even in the reference list. This shapes up to be an attempt of plagiarism.
A) First of all, I'm sorry for the mention of self-plagiarism. I admit to my mistake of not taking this part into account during the initial thesis submission process. I apologize again. The parts mentioned in the introduction have been corrected, and the quotations for figures have been added to the reference list. We will be careful not to do this in the future.
2. Any substantial novelty can be inferred from the document. Analytical methods are well-known and documented in the literature. Also the replacement of the structural element with equivalent constraint seems not new, but highly reminiscent of the direct stiffness method, a basic approach for the structural mechanics, use also before the advent of computers.
A) In order to reply your 2nd comment that any substantial novelty can not be found, the following are prepared.
Firstly, we have proposed polynomials combining fixed and simple supports to satisfy boundary condition at junctions between each subsystem. We know that this approach has never been tried.
Secondly, Although Bhat [4] proposed a fixed and simple support function, the calculation was performed by applying it to a simple plate. In addition, the function proposed by Bhat does not satisfy the natural condition in higher order terms of 2nd or higher order.
In this study, in order to compensate for this problem, calculations were performed for the two cases mentioned below at the connection point and the results were compared in section 4.1.
Case 1) Displacement, slope, moment continuity (total sum of natural conditions is continuous)
Case 2) Displacement, slope continuity (ignoring natural conditions)
For reference, the geometrical boundary condition mentioned in this manuscript refers to the boundary condition for displacement and slope, and the natural boundary condition refers to the boundary condition for moment. [4]
Third, in order to confirm the usefulness of the proposed method, a numerical analysis was performed on the representative shape of two & three components typical.
In particular, for the two component type, various verifications were performed in the entire length range 1 according to the length ratio (LA : LB).
Fourth, frequently, only specific subcomponent is more concerned for vibration analysis. In this case, the suitable boundary conditions to consider the static and dynamic coupling from the other subcomponent through junctions should be provided. However, the suggestions for such boundary conditions are hardly found. In this study, in order to calculate the above case, A simplified method that can reduce the degree of freedom up to 50% by matching the subcomponent mode and the interest component mode as a constraint condition at the junctions is also proposed.
The purpose of the simplification method presented in this study is to show that it is possible to calculate a method that can reduce the degree of freedom by 50%, rather than a method for comparing numerical calculation results with the existing method. Although this method is somewhat excessive, as a result, it satisfies the FEA result and the analysis error of 15%, which is appropriate as an approximate numerical methodology, so it is considered to be efficient for approximate numerical calculation.
Fifth, a three component structure was used for the calculation of structures in which symmetric and asymmetric modes occur repeatedly. A mode function having an appropriate boundary condition for a three component structure is proposed. The case of three component structures, fixed-fixed, fixed-simple, simple-fixed, fixed-free, simple-simple, and displacement functions were used.
3. The document is poorly written. In the following a non-comprehensive list of detected problems is reported.
i. Improper paragraph subdivision of the document. Why most of paragraphs are just one sentence long?
A) For paragraphs of the document, I will ask the editors to avoid making any possible modifications.
ii. geometrical conditions and natural conditions at junctions (Line 13) repeated word conditions
A) The pointed out was deleted as it seemed to interfere with understanding of this manuscript, and the text was revised.
iii. Polynomials (Line 14) wrong use of capital letter
A) Corrected capital letter.
iv. each shipyard has developed and used an approximate vibration calculation program according to the circumstances of the design. (Line 29) vague statement, it is more reasonable to believe that practitioners use commercial software nowadays. Provide suitable references.
A) We have added use cases of related programs to the references.
v. the beam function is used in the assumed mode function (Line 42) what is the meaning of this sentence?
A) I decided that it was not helpful to understand the content, so I deleted the pointed out and modified it as follows.
Before) when the analytical method is applied, the beam function is used in the assumed mode function.
Revision) when the analytical method is applied, the Euler’s beam function is used.
vi. This method is an introduction to a methodology (Line 72) tautology.
A) Corrected a mistake part of the original manuscript.
vii. The words coordinate and variable have different meaning. Conversely authors use them as synonymous e.g., if wa is a coordinate (Figure 2), it can not be also the displacement variable (Equations (1) and (2)).
A) In the revised manuscript, the subscripts are summarized and shown. Wa mentioned in formulas (1) and (2) also distinguished the displacement function and the total deflection equation into capital and small letters.
viii. Figure 6, wrong representation of the symmetry constraints.
A) What you pointed out is correct. If the calculation is performed using a fixed boundary condition in the middle, it is within the allowable range, but as you pointed out, in the case of symmetry, deflection occurs at the intermediate point, so I think it is wrong to express it as if it is an actual fixed condition. Corrected the picture.
ix. Section 2.2 looks like the copy paste of Section 2.1. It just consider 3 segments instead of 2.
A) In this paper, a numerical calculation was performed by introducing two and three components, which are typical structures, to explain the basic concept of the boundary condition of the junction and to confirm the usefulness of the proposed method.
In particular, for the two component type, various verifications were performed in the entire length range 1 according to the length ratio (LA : LB).
Round 2
Reviewer 1 Report
The authors have adequately addressed the comments.
The only concern is that the first paragraph of the introduction is very similar to another paper, 'Normal Mode Analysis for Connected Plate Structure Using Efficient Mode Polynomials with Component Mode Synthesis,' written by the same authors. (https://www.mdpi.com/2076-3417/10/21/7717/html?)
Author Response
The authors have adequately addressed the comments.
The only concern is that the first paragraph of the introduction is very similar to another paper, 'Normal Mode Analysis for Connected Plate Structure Using Efficient Mode Polynomials with Component Mode Synthesis,' written by the same authors. (https://www.mdpi.com/2076-3417/10/21/7717/html?)
A) First of all, I'm sorry, I agree with the opinion that they are very similar. The revised manuscript has been modified similar to the manuscript mentioned without compromising the need for research. I apologize again.
Reviewer 3 Report
The novelty of the paper is modest. Other well established techniques will provide similar or better results.
Author Response
The novelty of the paper is modest. Other well established techniques will provide similar or better results.
A) Firstly, In the calculation of component mode synthesis, since the junction boundary condition is neither a fixed nor a simple condition, the question of how to express this was motivated by the research. And, I tried to express it with a combination of two conditions, fixed and simple, and it was confirmed that it was very similar to the FEM result.
Secondly, when you are interested in one member of the connection structure, we proposed a method to simplify the calculation process by assigning an appropriate boundary condition considering the dynamic and static coupling of the other member.
Currently, I have been doing vibration work in a shipyard for 15 years. Recently, there are times when it is necessary to quickly verify in practice, and this method is usefully utilized.
As far as I know, these two methods have not yet been published. So I was very happy when I confirmed that the results of the two proposed methods are similar to those of the FEM.
Please, review the motives of my research and ask for your positive comments.